# Behavioral Changes in Preschool- and School-Age Korean Children: A Network Analysis

**DOI:** 10.3390/children9050677

**Published:** 2022-05-06

**Authors:** Eun-Kyoung Goh, Hyo Jeong Jeon

**Affiliations:** 1Human Life Research Center, Dong-A University, Busan 49315, Korea; ekgoh72@dau.ac.kr; 2Department of Child Studies, College of Humanities, Dong-A University, Busan 49315, Korea

**Keywords:** CBCL, preschool- and school-age, network analysis

## Abstract

The relationships between symptoms that comprise behavioral problems in children can be traced longitudinally to provide long-term support. This study identified signs that should be considered important in school age children by tracking changes in the relationships between different symptoms of behavioral problems in preschool and school age children. This study used Gaussian graphical network analysis to clarify the interaction of the overall subscales constituting the K-CBCL (Korean Child Behavior Checklist) and centrality in the network. In the Panel Study on Korean Children (PSKC), the K-CBCL/1.5–5 was used for children up to age six (first grade, elementary school), and the K-CBCL/6–18 was used for older children. In this study, 1323 PSKC samples (boys, *n* = 671; girls, *n* = 652) were used to distinguish nonclinical and (sub)clinical groups (T-score ≥ 60) compared to node centrality in each group’s CBCL subscale networks. Depression/anxiety was a persistent core symptom of the behavioral problem network in 5- and 7-year-old children. A new core symptom in 7-year-old children was posttraumatic stress problems added in version CBCL/6-18. Based on these results, it is necessary to consider both anxiety/depression and posttraumatic stress problems in preschool children to support the adaptation of school-age children.

## 1. Introduction

Korean government agencies revised the standardized Child Behavior Checklist (CBCL) into a Korean version to use mainly as a tool to measure the problem behaviors of children and adolescents. According to the survey conducted by the Korean Ministry of Health and Welfare, using the Korean Child Behavior Checklist for ages 6–18 (K-CBCL/6–18) version, the problem behaviors of Korean children are increasing, and continuous investigation and support are needed [1]. Behavioral problems in very young Korean children negatively affect school adjustment, even after entering elementary school [2]. To support this prophylactically, it is necessary to investigate the process by which preschool-age problem behaviors develop into school-age problems behaviors. A previous study reported an average difference between externalized problem behaviors in children in early childhood and elementary-school students in Korea [3]. However, it is difficult to understand the reciprocity between the sub-symptoms constituting the problem behavior by simply comparing the averages of the problem behavior, and it is difficult to accordingly determine preventive support. Therefore, it is necessary to compare the interactions between the sub-symptoms of problem behaviors over time.

The Korea Institute of Child Care and Education has measured behavioral problems by using the K-CBCL/1.5–5 for ages 4–6 [4] and the K-CBCL/6–18 for ages 7–9 [5] from the Panel Study on Korean Children (PSKC) born in 2008 and continues to track them (PSKC). There are only nine criteria common to Korean tools for preschoolers and school-age children, and the K-CBCL/6–18 version has been expanded to address more behavioral problem criteria than the 1.5–5 version. 

It has been reported that the CBCL is a reliable tool and that assessments measured at a young age tend to be maintained in later longitudinal studies [6,7]. However, it has also been reported that the stability of CBCL results depends on the categories of internalization and externalization [8], age [9], and clinical group status [10]. This assumes that behavioral problems are differentiated in various ways as children develop. Therefore, it is necessary to analyze whether the core symptoms of behavioral problems change if the CBCL measurement tool changes as children transition from preschool to school age. 

If child behavioral problems are applied to Gaussian graphical networks (GGNs), it can be understood that behavioral problem symptoms are structurally connected and interact in a complex manner. In this regard, depression and anxiety symptoms in early childhood are not distinguished as independent problem behaviors [11,12,13,14,15,16,17]. Applying these psychological relationships to network analysis methods can reveal how interactions between psychological symptoms develop [18].

In addition, scholars have advised using a person-oriented approach focused on group characteristics, rather than relationships between variables, to analyze child mental development [19,20]. If behavioral problems persist for a long time [7,21], a longitudinal study is needed to track the characteristics of these groups; however, this approach remains insufficient.

This study sought to determine how the relationships between behavioral problem symptoms change as a child develops. The specific goal was to confirm whether the relative influence of behavioral problem criteria differed depending on the (sub)clinical criteria of CBCL total problem behavior at each time point in early childhood and school age. However, this study did not deduce causal relationships between CBCL subscales. Therefore, we used a network analysis method to track the same subject by investigating whether significant symptoms changed in the preschool- and school-age behavioral problem symptom networks. We also examined the change in the proportion of the (sub)clinical population during the transition from early childhood to school age. In sum, this study offers insights into best practices for long-term support for problem behaviors in childhood by exploring whether the characteristics of each developmental stage can be identified according to the clinical group standards of problem behaviors. In order to compare the behavioral problems between preschool and school age, data measured at 5 and 7 years were used, considering the time when different versions of the tool were used and the Korean school system. This study was activated by three research questions: What are the core symptoms in the network of behavioral problem symptoms at 5 and 7 years of age?What are the core symptoms of the nonclinical and (sub)clinical groups in the behavioral problem symptom network at 5 and 7 years of age?How does the transition of core symptoms between the nonclinical and (sub)clinical groups change from 5 to 7 years of age?

## 2. Materials and Methods

### 2.1. Sample

The PSKC [22] are panel data that continuously track the development data of Korean children born in 2008, from the newborn period to the present (latest year 2021). Researchers can download and analyze the panel data simply by entering their affiliations. CBCL data collected in the PSKC in 2012 (w5), 2013 (w6), and 2014 (w7) were measured with a 1.5–5-year version. In addition, the data collected in 2015 (w8) and 2017 (w10) used a 6–18-year version. In PSKC, which tracks the development of a total of 2150 newborns, for the first time at w5, 1694 children were measured for behavioral problems with the CBCL, and at w10, 1461 children were measured. The sample used for network analysis in this study was data from 1323 children whose CBCL values were not omitted from w5 to w10. The CBCL data were obtained by calculating the raw score and T-score of the total behavioral problem at the publishing company that verified the Korean version of the tool and supplied the manual. To compare the networks of behavioral problem symptoms in the (sub)clinical and nonclinical groups, this study the classified the two groups and compared them based on the (sub)clinical reference value (T-score ≥ 60). The borderline of all (sub)clinical groups from w5 to w10 was T-score = 60. At the age of 5 years (w6), the sample was divided into a (sub)clinical group (*n* = 146) and a nonclinical group (*n* = 1177), and at the age of 7 years (w8), the sample was divided into a (sub)clinical group (*n* = 171) and a nonclinical group (*n* = 1152). Regarding the demographics of the study participants, the frequencies of gender and household income are similar. Additionally, most parents had a college degree or higher (Table 1). Among 1323 children, language disorders and brain lesions (*n* = 4, 0.3%) were included, but there were no children with mental disorders, such as ADHD.

### 2.2. Measures

#### CBCL

In this study, the same CBCL behavior problems were encoded from CB1 to CB22 so that they could be used with the same code in both versions. The subscales of K-CBCL/1.5–5 [4] include emotionally reactive (CB1), anxious/depressed (CB2), somatic complaints (CB3), and withdrawn (CB4), which correspond to internalizing problems. Externalizing problems include sleep problems (CB5), attention problems (CB6), aggressive behaviors (CB7), and other problems (CB8). Internalizing and externalizing behavioral problems constitute a number of problems. In addition, including DSM affective problems (CB9), DSM anxiety problems (CB10), DSM pervasive developmental problems (CB11), DSM attention-deficit/hyper-activity problems (CB12), and DSM oppositional defiant problems (CB13), following the DSM diagnostic criteria, at age 5, the total number of nodes in the behavioral symptom network was 13.

Sub-symptoms of K-CBCL/6–18 [5] include anxious/depressed (CB2), somatic complaints (CB3), and withdrawn/depressed (CB14), which are internalizing problems. In addition, rule-breaking behavior (CB15) and aggressive behaviors (CB7) correspond to externalizing problems. Attention problems (CB6), other problems (CB8), social problems (CB16), and thought problems (CB17) are all included in the total problems, along with internalization and externalization problems. Additionally, there are DSM affective problems (CB9), DSM anxiety problems (CB10), DSM attention-deficit/hyper-activity problems (CB12), DSM oppositional defiant problems (CB13), DSM somatic problems (CB18), and DSM conduct problems (CB19) that follow the DSM diagnostic criteria. Including the obsessive–compulsive symptom (CB20), posttraumatic stress problems (CB21), and sluggish cognitive tempo (CB22) corresponding to the special scale, the total number of nodes constituting the behavioral problem network at the age of 7 years is 18. Both versions of the CBCL data were measured by parents or guardians. In this sample (*n* = 1323), the reliability of the two versions of the CBCL overall scale used at 5 and 7 years was 0.95 (K-CBCL/1.5–5) and 0.94 (K-CBCL/6–18). 

### 2.3. Data Analysis

#### 2.3.1. Gaussian Graphical Network Analysis

GGN is a methodology used to analyze the expected influence of a specific phenomenon by identifying the centrality of a specific node in the network structure. The expected influence is analyzed through node strength, closeness, betweenness, and weight, which are calculated by specific nodes and edges [24]. Node strength is a calculation of how directly a node is connected to other nodes; that is, the absolute sum of the node’s correlations with other nodes. Closeness refers to whether a node is indirectly connected to other nodes; that is, the average length of the edges is calculated. Betweenness is a calculation of the importance of a node in the shortest path between two other nodes; that is, the number of times the node is included in the shortest path. The weight is a value calculated from the node strength, closeness, and betweenness of the predicted influence of the node in the network.

First, we analyzed the network of all children’s data collected for each wave of the PSKC. Next, we compared the network structures of the nonclinical group and the (sub)clinical group in the data of w6 and w8 with different CBCL norms. We compared the interconnectivity of behavioral problems by examining the expected influence of nodes in the networks of the w6 and w8 data. Finally, by testing the robustness of each network, a network showing stability was selected. Then, the network difference between the nonclinical group and the (sub)clinical group was estimated, and node predictability was compared.

#### 2.3.2. Network Estimation and Visualization

In the network image of this study, the nodes are the evaluation values of behavioral problems measured by the CBCL subscale. The network of behavioral problems measured up to age 6 consists of 13 nodes, and the network measured after age 7 consists of 18 nodes. The edge connecting the nodes in the network graphic image reflects the partial correlation value between the two behavioral symptoms. In addition, the edge does not assume causality between nodes, creating undirected networks with partial correlation estimates. In this study, networks were created according to the child’s age and clinical criteria. For this network modeling, the force-directed Fruchterman–Reingold algorithm [25,26] of the R-package *qgraph* [24] was used.

#### 2.3.3. The Network Comparison Test

The network comparison test (NCT) compares the network structure, global strength, and edge strength in two groups of networks. We compared the network differences between the nonclinical and (sub)clinical groups at 5 and 7 years. The NCT used the R-package *network comparison test* [27] in R statistical software version 4.0.5. 

#### 2.3.4. Node Predictability

The predictability of network nodes is calculated as the percentage of shared variance with the surrounding nodes [28]. In this study, the predictability of a node indicates how well the behavioral problem that the node represents can be predicted by other behavioral problems.

#### 2.3.5. Transition Analysis

A transition analysis was performed to determine how the nonclinical and (sub)clinical groups at w6 transferred to the nonclinical and (sub)clinical groups at w8. The proportion of children belonging to the nonclinical and (sub)clinical groups changed during the transition from 5 to 7 years. In addition, we selected behavioral problems with a relatively high expected influence from the networks of four groups (expected influence ≥ 0.5) and checked the T-scores of behavioral problems that changed in both periods.

## 3. Results

### 3.1. Network Inference by Age

#### 3.1.1. Sample Characteristics

The difference in total problem T-scores between the nonclinical and (sub)clinical groups (T-score 60) was significant at all times (*p* < 0.001). The mean of the total problems of the nonclinical group measured using the CBCL/1.5–5 version was highest at the time of the first measurement and decreased afterward. The mean of the total problems of the nonclinical group measured after changing to the CBCL/6–18 version was also highest when measured and decreased afterward. However, the means of the (sub)clinical groups for each period did not significantly differ (Figure 1).

#### 3.1.2. Network Stability

We used the *CS* coefficient to evaluate the stability of the network. If the *CS* coefficient is above 0.5, the network is considered stable; when the coefficient is higher than 0.7, the network is considered to have strong stability [29]. In the network of all samples for each wave from w5 to w10, the networks with *CS* coefficients greater than 0.5 were w6 and w8: w5 (*CS* = 0.44), w6 (*CS* = 0.51), w7 (*CS* = 0.44), w8 (*CS* = 0.52), and w10 (*CS* = 0.44). Therefore, in the stable networks of w6 and w8, the networks of the nonclinical and (sub)clinical groups were compared again. In w6, the network stability in both nonclinical networks (*CS* = 0.52) and (sub)clinical networks (*CS* = 0.51) was clear. In w8, the nonclinical network (*CS* = 0.75) showed strong stability and the (sub)clinical network (*CS* = 0.52) showed clear stability.

#### 3.1.3. Replicability: Node Centrality in w6 vs. w8 Networks

Among the non-zero edges constituting the network at w6, four pairs showed a strong positive correlation of more than 0.5 (*p* < 0.001), and the association was greater than that of other combinations of behavioral problems: anxious/depressed and DSM anxiety problems (CB2: CB10), withdrawn and DSM pervasive developmental problems (CB4:CB11), attention problems and DSM attention-deficit/hyper-activity problems (CB6:CB12), and aggressive behaviors and DSM oppositional defiant problems (CB7:CB13). At 7 years (w8), three pairs showed a strong positive correlation above 0.5 (*p* < 0.001): attention problems and social problems (CB6:CB16), DSM affective problems and thought problems (CB9:CB17), and DSM anxiety problems and DSM conduct problems (CB10:CB19) (Figure 2a).

At 5 years (w6), there were six central nodes with an *EI* (expected influence) greater than 0.5 in the network of the entire sample: DSM anxiety problems (CB10), anxious/depressed (CB2), DSM affective problems (CB9), DSM pervasive developmental problems (CB11), emotionally reactive (CB1), and aggressive behaviors (CB7). On the other hand, the node with the lowest centrality in the network at 7 years (w8) was somatic complaints (CB3). At 7 years (w8), the central nodes (*EI* > 0.5) in the network were posttraumatic stress problems (CB21), DSM conduct problems (CB19), DSM attention-deficit/hyper-activity problems (CB12), DSM affective problems (CB9), anxious/depressed (CB2), and attention problems (CB6). The node with the lowest centrality in the w8 network was the DSM oppositional defiant problem (CB13).

The commonly central nodes at 5 years and 7 years were anxious/depressed (CB2) and DSM affective problems (CB9); the node with a weak centrality was other problems (CB8). The more central nodes in the network at 5 years compared to 7 years were aggressive behaviors (CB7) and DSM anxiety problems (CB10). Meanwhile, the more central nodes in the network at 7 years compared with 5 years were attention problems (CB6) and DSM attention-deficit/hyper-activity problems (CB12) (Figure 2b).

### 3.2. Network Inference According to Clinical Risk

#### 3.2.1. Node Centrality in Nonclinical vs. (Sub)Clinical Networks at w6

In both the w6 nonclinical and (sub)clinical group networks, four pairs showed a strong positive correlation above 0.5 (*p* < 0.001): anxious/depressed and DSM anxiety problems (CB2:CB10), withdrawn and DSM pervasive developmental problems (CB4:CB11), attention problems and DSM attention-deficit/hyper-activity problems (CB6:CB12), and aggressive behaviors and DSM oppositional defiant problems (CB7:CB13). However, the number of non-zero edges in the two groups was different. The strongest positive correlation in the nonclinical group network was attention problems and DSM attention-deficit/hyper-activity problems (CB6:CB12); meanwhile, the strongest positive correlation in the (sub)clinical group network was aggressive behaviors and DSM oppositional defiant problems (CB7:CB13) (Figure 3a).

At 5 years (w6), the central nodes (*EI* > 0.5) in the (sub)clinical group network were DSM anxiety problems (CB10), anxious/depressed (CB2), DSM affective problems (CB9), DSM pervasive developmental problems (CB11), emotionally reactive (CB1), and aggressive behaviors (CB7). The lowest centrality node was somatic complaints (CB3). The characteristics of this network were similar to those of the entire group network at w6.

The central nodes (*EI* > 0.5) in the nonclinical group network were DSM attention-deficit/hyper-activity problems (CB12), DSM pervasive developmental problems (CB11), DSM affective problems (CB9), and emotionally reactive (CB1). The lowest centrality node was sleep problems (CB5). 

At 5 years (w6), the most central node in the nonclinical group network was the DSM attention-deficit/hyper-activity problems (CB12); the most central node in the (sub)clinical group network was the DSM anxiety problem (CB10). The commonly weak centrality nodes in the nonclinical and (sub)clinical groups were somatic complaints (CB3) and sleep problems (CB5) (Figure 3b).

#### 3.2.2. Node Centrality in Nonclinical vs. (Sub)Clinical Networks at Wave 8

In both the w8 nonclinical and (sub)clinical group networks, four pairs showed a strong positive correlation above 0.5 (*p* < 0.001): anxious/depressed and rule-breaking behavior (CB2:CB15), attention problems and social problems (CB6:CB16), DSM anxiety problems and DSM conduct problems (CB10:CB19), and DSM attention-deficit/hyper-activity problems and DSM somatic problems (CB12:CB18). Meanwhile, in the (sub)clinical group, a positive correlation above 0.5 was found in three pairs (*p* < 0.001): attention problems and social problems (CB6:CB16), DSM affective problems and thought problems (CB9:CB17), and DSM anxiety problems and DSM conduct problems (CB10:CB19). In the (sub)clinical group, a strong positive correlation was found in an additional three pairs. In both groups, attention problems and social problems (CB6:CB16) had the greatest correlation (Figure 4a).

At 7 years (w8), the central nodes (*EI* > 0.5) in the (sub)clinical group network were posttraumatic stress problems (CB21), DSM conduct problems (CB19), DSM attention-deficit/hyper-activity problems (CB12), DSM affective problems (CB9), anxious/depressed (CB2), and attention problems (CB6). The lowest centrality node was the DSM oppositional defiant problem (CB13). The characteristics of this network were similar to those of the entire group network at w6. 

The central nodes (*EI* > 0.5) in the nonclinical group network were anxious/depressed (CB2), DSM attention-deficit/hyper-activity problems (CB12), DSM affective problems (CB9), posttraumatic stress problems (CB21), and attention problems (CB6). The lowest centrality node was the sluggish cognitive tempo (CB22).

At 7 years (w8), the core node in the nonclinical group network was anxious/depressed (CB2) and in the (sub)clinical group network it was posttraumatic stress problems (CB21).

However, in both networks, the nodes of posttraumatic stress problems (CB21), DSM attention-deficit/hyper-activity problems (CB12), DSM affective problems (CB9), anxious/depressed (CB2), and attention problems (CB6) all demonstrated high centrality. Meanwhile, the nodes with weak centrality in both networks were DSM oppositional defiant problems (CB13) and sluggish cognitive tempo (CB22) (Figure 4b).

#### 3.2.3. Mean Difference of Sub-Factors of CBCL by Groups

The difference in T-scores between the nonclinical and (sub)clinical groups for each behavioral problem was significant (*p* < 0.001). As for the effect size on the difference between the nonclinical group and the (sub)clinical group, the category of other problems (CB8) was the largest at 5 years (w6), but DSM attention-deficit/hyper-activity problems (CB12) was the largest at 7 years (w8) (Table 2). The prominent problem behaviors in the (sub)clinical group at age 5 were other problems, anxious/depressed, and emotionally reactive, while, at age 7, they were DSM attention-deficit/hyper-activity problems, aggressive behaviors, and posttraumatic stress problems. The problem behaviors that stand out in the (sub)clinical groups are similar to the central nodes of each network, except for other problems at age 5.

However, the difference in expected influences (*EI*) between the two groups was greatest in DSM attention-deficit/hyper-activity problems (CB12) at 5 years (w6) (Figure 2) and the largest in the obsessive–compulsive symptoms (CB20) at 7 years (w8) (Figure 3). Meanwhile, comparing the mean edge weights for each network, we see that the connectivity of nodes in the network of the (sub)clinical group was stronger than that of the nonclinical group at both 5 years (w6) and 7 years (w8) (Table 2).

#### 3.2.4. Node Predictability

The mean node predictability of the CBCL network was higher at 7 years than at 5 years. The mean node predictability of the 5-year dataset was 59.1%, while that of the 7-year network was 66.4%. In both periods, the mean node predictability of the (sub)clinical group was slightly higher than that of the nonclinical group. In the dataset of the 5-year nonclinical group, on average, 52.2% of the variance of each node was explained by the neighboring node, and in the (sub)clinical group, 58.9% was explained by the neighboring node. Even after 7 years, 57.2% of the variance of each node on average in the dataset of the nonclinical group was explained by the adjacent node, but in the (sub)clinical group, 66.6% is explained by the adjacent node.

At 5 years (w6), the most unpredictable node among the CBCL network nodes was sleep problems (CB5), which, on average, shared only 30.6% of the variance with surrounding nodes in the nonclinical group and only 33.8% in the (sub)clinical group. Meanwhile, regarding the most predictable nodes, aggressive behaviors (CB7) shared 70.8% of the variance with surrounding nodes on average in the nonclinical group, and DSM anxiety problems (CB10) shared 78.6% of the variance in the (sub)clinical group (Figure 5). The node that was the most difficult to predict among the CBCL network nodes at 7 years (w8) was DSM oppositional defiant problems (CB13), which was common to both the nonclinical group (28.1%) and the (sub)clinical group (36.2%). The most predictable node was DSM affective problems (CB9), which was common to both the nonclinical group (79.0%) and the (sub)clinical group (84.3%) (Figure 6).

At 5 years (w6), the most predictable node in the nonclinical group was CB7 (aggressive behaviors). It shared 70.8% of the variance with neighboring nodes on an average and interacted more with CB12 (DSM attention-deficit/hyper-activity problems) and CB13 (DSM oppositional defiant problems) in particular. In the (sub)clinical group, the most predictable node was CB10 (DSM anxiety problems), which, on an average, shared 78.6% of the variance with neighboring nodes and interacted more with CB2 (anxious/depressed) and CB5 (sleep problems) in particular. In the 7th year(w8), the most predictable node for both nonclinical and (sub)clinical groups was CB9 (DSM affective problems), which interacted more with CB17 (thought problems) and CB22 (sluggish cognitive tempo) in particular. However, the predictability was greater in the (sub)clinical group (84.3%) than in the nonclinical group (79.0%). 

#### 3.2.5. Network Comparison

##### Nonclinical vs. (Sub)Clinical at w6 

At 5 years (w6), there was no significant difference between the (sub)clinical group (*n* = 146) network and the nonclinical group (*n* = 1177); that is, the expected influence of the core node was maintained even when the network sample was randomly changed (global expected influence, *S* = 0.726, *p* = 0.867). Nonclinical groups were randomly sampled, and sample sizes were balanced in comparison to solve the instability of NCT that occurs when comparing groups with different sample sizes; however, no significant difference was observed in the global expected influences of the (sub)clinical group network (*S* = 0.669, *p* = 0.070). Therefore, it is difficult to see whether there is a clear difference in the centrality of the network nodes of the nonclinical group and the (sub)clinical group of behavioral problems.

##### Nonclinical vs. (Sub)Clinical at w8 

A significant difference was observed when comparing the global expected influence of the network of the (sub)clinical group (*n* = 171) at 7 years (w8) with the nonclinical group (*n* = 1152) (global expected influence, *S* = 1.652, *p* = 0.007). Even when comparing the same sample size (*n* = 171) with random sampling for the nonclinical group, the global expected influence of the (sub)clinical group network was significantly different (*S* = 1.652, *p* = 0.010).

In addition, the expected influences of the edge in the network of the (sub)clinical group were further reduced between aggressive behaviors, anxious/depressed (CB7: CB14, *p* = 0.027), DSM attention-deficit/hyper-activity problems, and DSM somatic problems (CB12: CB18, *p* = 0.009) compared to the network of the nonclinical group (Figure 3). Therefore, there is a difference in centrality in the network of the nonclinical group and the (sub)clinical group at 7 years.

### 3.3. Group Transition from w6 to w8

Focusing on six important problem behaviors (CB1, CB2, CB7, CB9, CB10, and CB11) with relatively high expected influences (*EI* > 0.5) of the 5-year dataset and six important problem behaviors (CB2, CB6, CB9, CB12, CB19, and CB21) of the 7-year dataset, we analyzed how the behavioral problems of the nonclinical group and the (sub)clinical group were transferred from 5 to 7 years. Notably, 50% of children in the (sub)clinical group at 5 years transferred to the (sub)clinical group, and the remaining 50% transferred to the nonclinical group at 7 years. In the (sub)clinical group of the 7-year dataset, 42.69% of cases were transferred from the (sub)clinical group of the 5-year dataset and 57.30% of the cases were transferred from the 5-year nonclinical group.

Anxious/depressed (CB2) and DSM attention-deficit/hyper-activity problems (CB12), which had a high centrality in the (sub)clinical group at 5 years, still had a high T-score in the (sub)clinical group at 7 years. In the 7-year network, the T-score also increased for posttraumatic stress problems (CB21) with increased node centrality. This highlights the importance of CB21 in the total behavior problem. In other words, as young children enter elementary school, CB21 becomes an important element of the total behavior problem (Table 3 and Figure 7).

## 4. Discussion

Recently, the network analysis framework has been used to analyze the relationships between the symptoms of mental problems. For example, some studies have used network analyses to uncover the relationship between the reciprocity and centrality of depression and anxiety symptoms [18,30,31], the centrality of PTSD symptoms [32,33], and the interaction between trauma and psychotic symptoms [34]. However, the reciprocity or centrality between the symptoms of various behavioral problems in children, especially the analysis according to age or clinical group, has not been revealed. The purpose of this study was to provide a long-term approach for children’s adaptation by longitudinally tracking the core symptoms that change over time among particular age groups while analyzing the reciprocity between the various symptoms constituting the behavioral problem through the network analysis frame.

### 4.1. Stability of the CBCL Behavioral Problem Network

Although there are differences between cultures [35], the CBCL is an internationally recognized tool for measuring behavioral problems in children [36]. This study, which compared groups of networks, is based on PSKC data, which measured CBCL at five time periods from 4 to 9 years of age. A guideline for network stability is that the *CS* coefficient of each GGN model should preferably be above 0.5; this was met when the children were aged 5 and 7, among the networks of both the nonclinical group and the (sub)clinical group [24,29]. Therefore, based on the network approach, we were able to analyze the interconnectivity between behavioral problems longitudinally according to clinical criteria. The results of our analysis support the idea that the network approach may explain the co-occurrence of mental disorders [37].

### 4.2. Core Behavioral Problems in CBCL Networks

First, looking at the pairs of nodes with high correlations, CBCL symptoms related to behavioral problems following the DSM criteria were mainly paired at 5 years. Naturally, behavioral problems following the DSM guidelines are highly correlated with similar symptoms [38]. However, at 7 years, strong positive correlations were found between attention problems and social problems, DSM affective problems and thought problems, and DSM anxiety problems and DSM conduct problems. As the number of behavioral problem symptoms increased from 13 to 18, strong correlations were also found between different behavioral problem symptoms, which suggests comorbid mental problems [39].

Second, the core behavioral problems of the whole sample by age are the same as the core behavioral problems of the clinically risky group. At age 5, six relatively important behavioral problems include being emotionally reactive, anxious/depressed, or aggressive; DSM affective problems; DSM anxiety problems; and DSM pervasive developmental problems. At age 7, six behavioral problems are central: being anxious/depressed, attention problems, DSM affective problems, DSM attention-deficit/hyper-activity problems, DSM conduct problems, and posttraumatic stress problems. 

Anxiety/depression is central at age 5 and remains central at age 7 as well. In previous studies, anxiety and depressive symptoms not only had a strong correlation with each other [18,40] but also with other behavioral problems. Prevalence of social anxiety and depression in childhood increases until adulthood [41], and the destructive and internalized behavioral problems of childhood lead to symptoms of anxiety in adulthood as well [18]. Among various behavioral problems, the importance of anxiety and depressive symptoms is maintained through the homogeneous or heterozygous transition from childhood to adulthood. 

At age 7, posttraumatic stress problems and attention-deficit/hyper-activity problems appear to be central symptoms. These results echo the findings of a study following a child from age 4 to age 10 that reported that early behavioral symptoms influence subsequent ADHD symptoms [42]. Similarly, a longitudinal study in Korea reported that the level of behavioral problems associated with posttraumatic stress problems (as per DSM guidelines) increases with grade level from ages 6 to 18 [43].

In addition, the high centrality of posttraumatic stress problems at 7 years suggests previous trauma exposure. The posttraumatic stress problems have a strong direct and indirect relationship with other symptoms and are longitudinally transferred to other disorders [32]; it is necessary to measure and prevent posttraumatic stress problems by following the DSM guidelines, even in early childhood [44]. The question of whether children and adolescents can develop PTSD has been controversial [45], but this view needs to be changed and taken seriously as a possible disorder in early childhood [46,47]. Among the symptoms of CBCL behavior problems, anxiety/depression symptoms persist as core symptoms at 5 and 7 years, but they also emerge as core symptoms at 7 years, along with attention-deficit/hyper-activity problems and posttraumatic stress problems. These results support the idea that various psychiatric symptoms have both continuity and discontinuity [48].

Meanwhile, DSM attention-deficit/hyper-activity problems at 5 years and obsessive–compulsive symptoms at 7 years showed a large gap in expected influences in networks between (sub)clinical and nonclinical groups. It is difficult to identify more prominent symptoms in the risk group when simply considering their comorbidity based on the difference in the level of behavioral problems between the two groups. At both the ages of 5 and 7, node connectivity in the network continued to be stronger in the (sub)clinical group than in the nonclinical group; therefore, the higher the risk, the more interconnectivity or comorbidity should be considered rather than individual symptoms alone.

### 4.3. Highly Predictable Behavioral Issues

It is easier to predict behavioral problems at 7 years than at 5 years in the (sub)clinical group than in the nonclinical group. The most predictable node in the (sub)clinical group’s network at 5 years was the DSM anxiety problem. The items measuring DSM anxiety problems in the CBCL are related to unstable attachment and separation anxiety in early childhood. Early in life, unstable attachment predicts both internalization [49,50] and externalization symptoms [51]. Considering the interconnectivity between behavioral problems, it will be easier to predict DSM anxiety problems related to an insecure attachment if information on various behavioral problems is given.

At age 7, DSM affective problems were the most predictable in the (sub)clinical group. The CBCL’s DSM affective problems are measured with items related to “feelings of inferiority and guilt”. If it becomes easier to predict children’s emotional problems from various behavioral problems as they reach school age, then the occurrence of various behavioral problems in Korean school-age children may be related to academic stress [52]. This high mean node predictability shows that it is very difficult to target specific symptoms, and it is necessary to collect and consider surrounding symptoms that may be related to target symptoms as much as possible for effective intervention [18,37,39]. 

### 4.4. Transfer of Core Behavioral Problems

Fifty percent of children in the (sub)clinical group at 5 years transition to the (sub)clinical group at 7 years. This supports the notion that problematic behaviors in childhood continue with advancing age [7,41,53]. Among the behavioral problems, anxiety/depression symptoms and DSM attention-deficit/hyper-activity problems, which were relatively central from 5 to 7 years, also maintained a high T-score. In addition, the T-scores of posttraumatic stress problems, which showed a newly high centrality at 7 years, were higher than other symptoms. Through the network approach, we confirmed that the importance of anxiety and depression was remarkably maintained from early childhood to early school age, and the importance of posttraumatic stress problems was newly emphasized at early school age. 

This study supports the idea that developmentally significant psychiatric symptoms in the network approach show longitudinally homotypic and heterotypic continuity phenomena [18,42]. In this sense, our study shows the usefulness of the network approach in understanding the developmental mechanisms of behavioral problems with increasing age.

### 4.5. Limitations and Suggestions

First, the data of this study were samples of general children, and the sample size of the clinical group was small. Inevitably, the cutoff criterion was lowered for comparison, and as a result, it was difficult to clearly see the characteristics of the clinical group. In the future, we propose a network study comparing a large-sized clinical group with a nonclinical group. 

Second, this study reported that behavioral problems such as anxious/depressed in preschoolers still became central behavioral problems in school age, and that posttraumatic stress problems emerged as a new important behavioral problem. However, the network approach cannot account for the influence of these behavioral problems. Future studies are needed to longitudinally track what factors affect developmental changes and transitions of behavioral problems.

Third, in our study, the accuracy of K-CBCL rating was limited in evaluating behavioral problems by relying only on the parent’s report. More accurately measured data from other sources or some additional measurements are needed for future study.

Finally, although very few children with developmental disorders were included in this study, the effects of disability could not be fully controlled. Future studies should consider the effects of somatic or psychiatric illness.

## 5. Conclusions

This study visually presented how the interconnectivity of CBCL behavioral problems in a network approach varies with time and clinical conditions. Through this, it effectively represented whether childhood psychiatric symptoms showed developmentally homotypic or heterotypic continuity. Anxious/depressed was a persistent core symptom of the behavioral problem network in 5- and 7-year-olds, and a new core symptom in 7-year-olds was a posttraumatic stress problems added in version CBCL6-18. Based on these results, it is necessary to consider both the anxious/depressed and posttraumatic stress problems issues of preschool children to support the adaptation of school-age children. In the future, longitudinal exploration and follow-ups are needed to elucidate the factors that reinforce core symptoms.

## Figures and Tables

**Figure 1 children-09-00677-f001:**
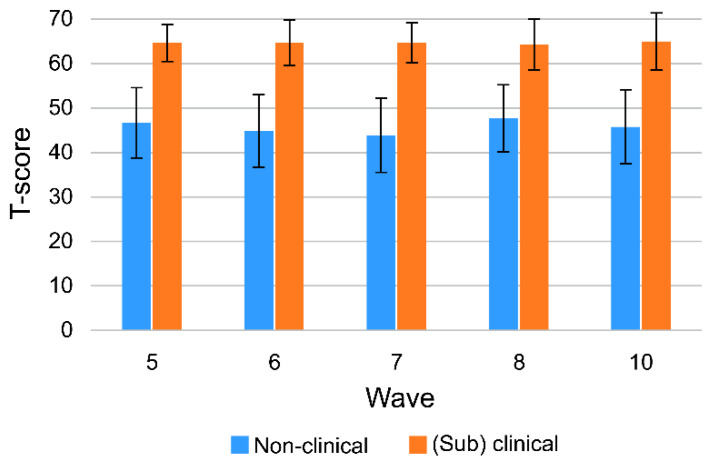
CBCL total problems (T-score).

**Figure 2 children-09-00677-f002:**
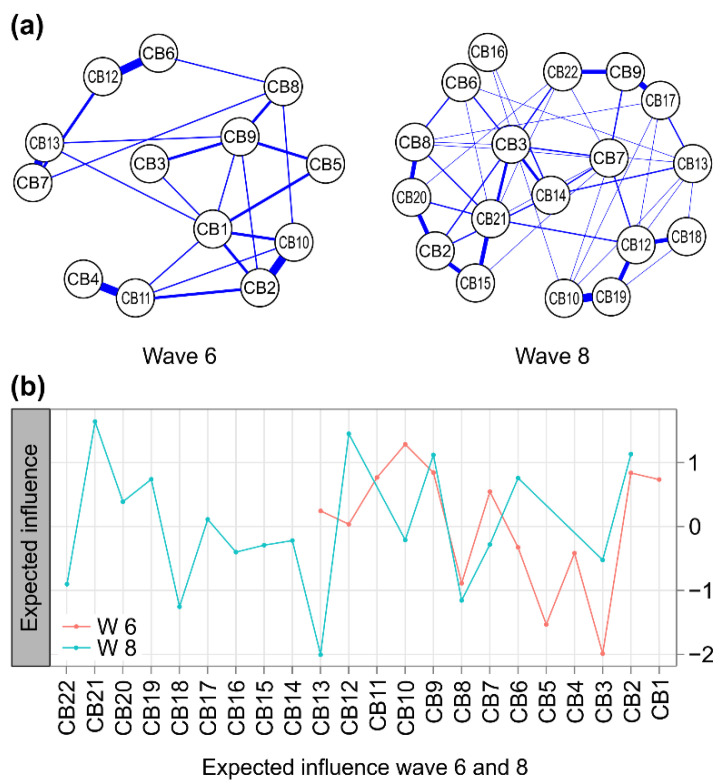
(**a**) Comparison of networks of 13 nodes with CBCL behavioral problems at 5 years (**left**) and 18 nodes at 7 years (**right**). The blue lines present positive associations, and the red lines indicate negative ones. The width and brightness of the edge indicate the strength of the association. The maximum edge of w6 is 0.64 (CB7 = aggressive behaviors; CB13 = DSM oppositional defiant problems), and the maximum edge of w8 is 0.73 (CB6 = attention problems; CB16 = social problems). (**b**) Individual node strength values are shown as standardized *z*-scores for w6 (orange) versus w8 (blue). CB1 = emotionally reactive; CB2 = anxious/depressed; CB3 = somatic complaints; CB4 = withdrawn; CB5 = sleep problems; CB6 = attention problems; CB7 = aggressive behaviors; CB8 = other problems; CB9 = DSM affective problems; CB10 = DSM anxiety problems; CB11 = DSM pervasive developmental problems; CB12 = DSM attention-deficit/hyper-activity problems; CB13 = DSM oppositional defiant problems; CB14 = withdrawn/depressed; CB15 = rule-breaking behavior; CB16 = social problems; CB17 = thought problems; CB18 = DSM somatic problems; CB19 = DSM conduct problems; CB20 = obsessive–compulsive symptom; CB21 = posttraumatic stress problems; CB22 = sluggish cognitive tempo.

**Figure 3 children-09-00677-f003:**
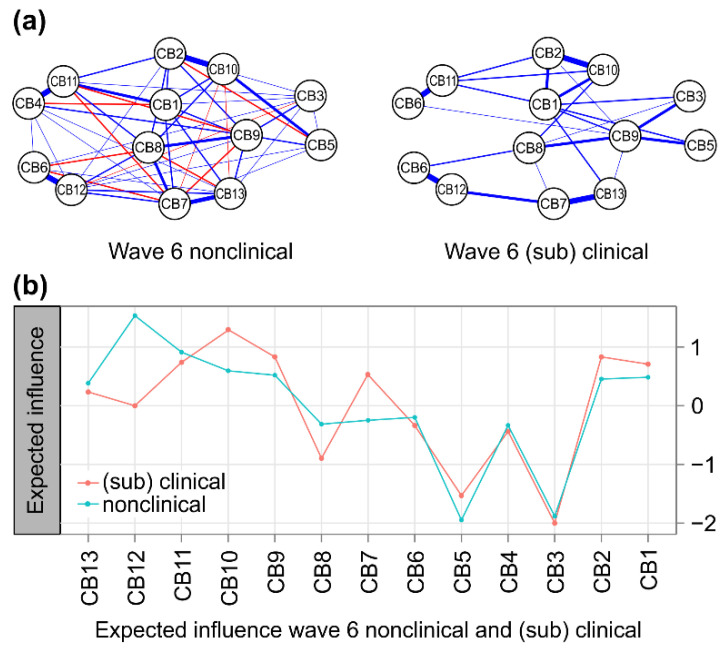
(**a**) A comparison of 13 nodes’ CBCL behavioral problems in nonclinical (**left**) and (sub)clinical (**right**) groups. The blue lines represent positive associations, and the red lines indicate negative ones. The width and brightness of the edge indicate the strength of the association. The maximum edge in the nonclinical group was 0.77 (CB6 = attention problems; CB12 = DSM attention-deficit/hyper-activity problems), and the maximum edge in the (sub)clinical group was 0.64 (CB7 = aggressive behaviors; CB13 = DSM oppositional defiant problems), but the maximum edge was unified as 0.55 for comparison. (**b**) Nonclinical (total problem behavior T-score < 60, blue) versus (sub)clinical (total problem behavior T-score ≥ 60, orange) individual node strength values are shown as standardized *z*-scores. CB1 = emotionally reactive; CB2 = anxious/depressed; CB3 = somatic complaints; CB4 = withdrawn; CB5 = sleep problems; CB6 = attention problems; CB7 = aggressive behaviors; CB8 = other problems; CB9 = DSM affective problems; CB10 = DSM anxiety problems; CB11 = DSM pervasive developmental problems; CB12 = DSM attention-deficit/hyper-activity problems and CB13 = DSM oppositional defiant problems.

**Figure 4 children-09-00677-f004:**
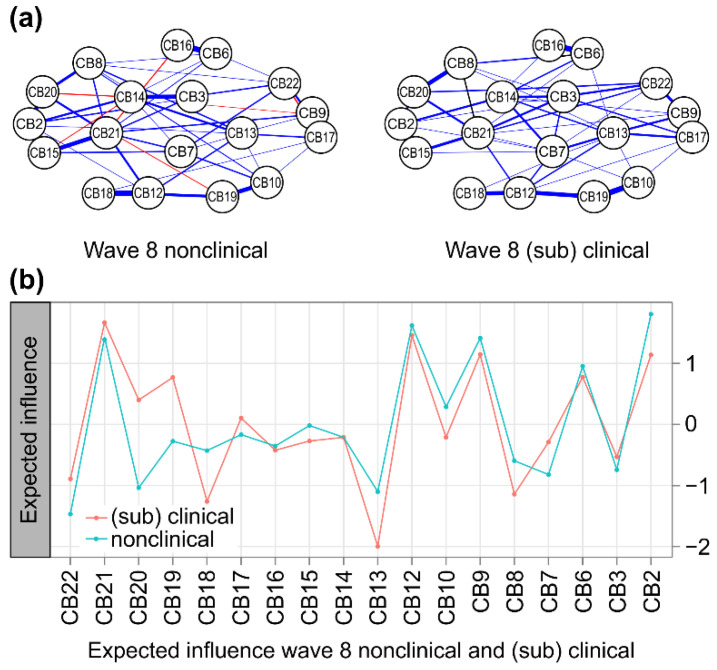
(**a**) Comparison of CBCL behavioral problems in nonclinical (**left**) and (sub)clinical (**right**) groups of 18 nodes. The blue lines present positive associations, and the red lines indicate negative ones. The width and brightness of the edge indicate the strength of the association. The maximum edge in the nonclinical group was 0.84 (CB6 = Attention problems; CB16 = Social Problems) and the maximum edge in the (sub)clinical group was 0.73 (CB6 = Attention problem; CB16 = Social Problems), but the maximum edge was unified as 0.55 for comparison. (**b**) Nonclinical (total problem behavior T-score < 60, blue) versus (sub)clinical (total problem behavior T-score ≥ 60, orange) individual node strength values are shown as standardized z-scores. CB1 = emotionally reactive; CB2 = anxious/depressed; CB3 = somatic complaints; CB4 = withdrawn; CB5 = sleep problems; CB6 = attention problems; CB7 = other problems; CB8 = other problems; CB9 = DSM affective problems; CB10 = DSM anxiety problems; CB11 = DSM pervasive developmental problems; CB12 = DSM attention-deficit/hyper-activity problems; CB13 = DSM oppositional defiant problems; CB14 = withdrawn/depressed; CB15 = rule-breaking behavior; CB16 = social problems; CB17 = thought problems; CB18 = DSM somatic problems; CB19 = DSM conduct problems; CB20 = obsessive–compulsive symptom; CB21 = posttraumatic stress problems; CB22 = sluggish cognitive tempo.

**Figure 5 children-09-00677-f005:**
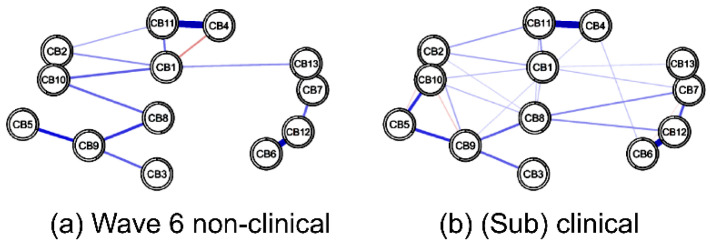
Comparison of the (**a**) nonclinical and (**b**) (sub)clinical CBCL 13-node networks. The shaded rings around each node indicate predictability (%).

**Figure 6 children-09-00677-f006:**
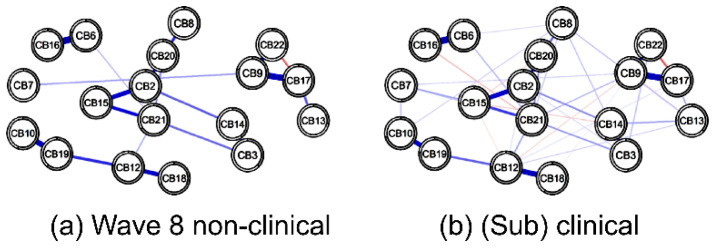
Comparison of CBCL 18-node networks for (**a**) nonclinical and (**b**) (sub)clinical groups. The shaded rings around each node indicate predictability (%). CB1 = emotionally reactive; CB2 = anxious/depressed; CB3 = somatic complaints; CB4 = withdrawn; CB5 = sleep problems; CB6 = attention problems; CB7 = aggressive behaviors; CB8 = other problems; CB9 = DSM affective problems; CB10 = DSM anxiety problems; CB11 = DSM pervasive developmental problems; CB12 = DSM attention-deficit/hyper-activity problems; CB13 = DSM oppositional defiant problems; CB14 = withdrawn/depressed; CB15 = rule-breaking behavior; CB16 = social problems; CB17 = thought problems; CB18 = DSM somatic problems; CB19 = DSM conduct problems; CB20 = obsessive–compulsive symptom; CB21 = posttraumatic stress problems and CB22 = sluggish cognitive tempo.

**Figure 7 children-09-00677-f007:**
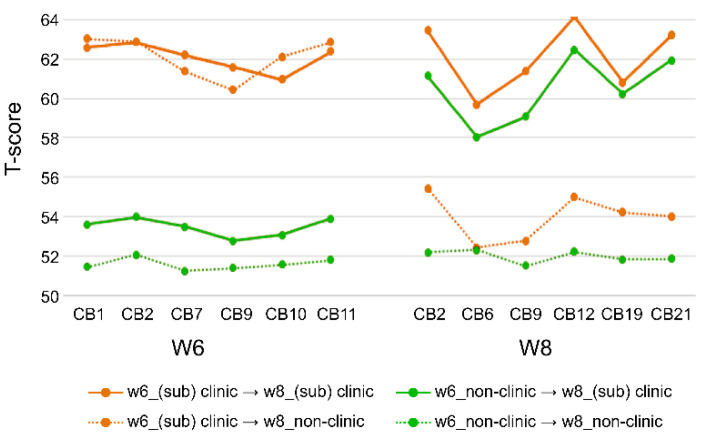
Approximately 50% of the (sub)clinical group at w6 was maintained as the (sub)clinical group at w8 (T-score ≥ 60). CB1 = emotionally reactive; CB2 = anxious/depressed; CB6 = attention problems; CB7 = aggressive behaviors; CB9 = DSM affective problems; CB10 = DSM anxiety problems; CB11 = DSM pervasive developmental problems; CB12 = DSM attention-deficit/hyper-activity problems; CB19 = DSM conduct problems; CB21 = posttraumatic stress problems.

**Table 1 children-09-00677-t001:** Participant demographics by sample (*n* = 1323).

	*M*	*SD*	*n*	%	Missing (%)
Monthly age	50.99	1.21			
Boys			671	50.7	0.0
Father with a college degree or higher			961	72.6	1.0
Mother with a college degree or higher			935	70.7	0.5
Households with above-average income in 2012 [23]			697	52.7	1.1

**Table 2 children-09-00677-t002:** Mean difference of CBCL behavior problems (T-score) between nonclinical and (sub)clinical groups and mean edge weight of network by groups.

Node Label	Norms	w6			w8		
		Nonclinical	(Sub)Clinical		Nonclinical	(Sub)Clinical	
		(*n* = 1177)	(*n* = 146)		(*n* = 1152)	(*n* = 171)	
		*M* (*SD*)		*t* (*g*)	*M* (*SD*)		*t* (*g*)
CB1	Emotionally reactive	51.64 (3.10)	62.81 (7.49)	−17.82 *** (2.91)			
CB2	Anxious/depressed	52.22 (3.72)	62.86 (7.09)	−17.83 *** (2.52)	52.41 (3.86)	62.13 (6.78)	−18.32 *** (2.24)
CB3	Somatic complaints	52.56 (4.38)	59.49 (7.40)	−11.08 *** (1.44)	52.35 (3.81)	60.05 (6.86)	−14.34 *** (1.78)
CB4	Withdrawn	52.21 (3.89)	61.66 (6.51)	−17.17 *** (2.22)			
CB5	Sleep problems	51.47 (3.50)	58.47 (6.94)	−12.01 *** (1.74)			
CB6	Attention problems	51.85 (3.68)	59.32 (6.61)	−13.39 *** (1.82)	52.32 (3.91)	58.75 (7.73)	−10.68 *** (1.40)
CB7	Aggressive behaviors	51.43 (3.03)	61.79 (5.48)	−22.43 *** (3.06)	52.61 (3.79)	62.75 (5.90)	−21.83 *** (2.46)
CB8	Other Problems	51.55 (3.05)	63.23 (5.68)	−24.40 *** (3.40)	52.76 (4.25)	61.96 (6.90)	−16.97 *** (1.97)
CB9	DSM affective problems	51.50 (3.20)	61.01 (6.71)	−16.89 *** (2.54)	51.60 (3.07)	60.06 (7.18)	−15.21 *** (2.20)
CB10	DSM anxiety problems	51.71 (3.33)	61.53 (7.50)	−15.64 *** (2.45)	53.85 (4.34)	61.57 (6.42)	−15.21 *** (1.66)
CB11	DSM pervasive developmental problems	51.96 (3.79)	62.62 (7.47)	−16.96 *** (2.45)			
CB12	DSM attention-deficit/hyper-activity problems	52.20 (3.82)	60.77 (6.77)	−15.00 *** (2.02)	52.39 (3.72)	63.19 (5.76)	−23.77 *** (2.43)
CB13	DSM oppositional defiant problems	51.63 (3.32)	60.93 (6.39)	−17.30 *** (2.46)	52.88 (4.46)	62.25 (6.82)	−17.42 *** (1.94)
CB14	Withdrawn/depressed				51.75 (3.54)	60.17 (6.99)	−15.46 *** (2.03)
CB15	Rule-breaking behavior				52.80 (4.40)	62.47 (8.58)	−14.46 *** (1.88)
CB16	Social problems				52.26 (4.42)	57.26 (8.15)	−7.85 *** (0.99)
CB17	Thought problems				51.79 (3.36)	60.38 (7.55)	−14.66 *** (2.07)
CB18	DSM somatic problems				52.52 (4.01)	61.14 (6.76)	−16.25 *** (1.93)
CB19	DSM conduct problems				51.98 (3.86)	60.47 (7.32)	−14.89 *** (1.90)
CB20	Obsessive–compulsive symptom				52.75 (4.32)	60.78 (7.36)	−13.94 *** (1.67)
CB21	Posttraumatic stress problems				51.98 (3.28)	62.49 (6.15)	−21.89 *** (2.79)
CB22	Sluggish cognitive tempo				51.17 (3.33)	58.32 (8.44)	−10.94 *** (1.65)
Mean Edge Weight		0.088	0.176		0.089	0.134	

*g* = Hedges’ *g* (effect size). *** *p* < 0.001.

**Table 3 children-09-00677-t003:** The transition of nonclinical and (sub)clinical groups to nodes with high expected influences.

		T-Score *M* (*SD*)			
Class	w6	(sub)clinical	(sub)clinical	nonclinical	nonclinical
	w8	(sub)clinical	nonclinical	(sub)clinical	nonclinical
	*n* (%)	73 (5.00)	73 (5.00)	98 (7.41)	1079 (81.56)
w6	CB1	62.58 (7.47)	63.04 (7.56)	53.59 (4.25)	51.46 (2.92)
	CB2	62.85 (7.25)	62.88 (6.98)	53.99 (4.38)	52.06 (3.62)
	CB7	62.21 (5.54)	61.38 (5.44)	53.50 (4.21)	51.24 (2.82)
	CB9	61.59 (7.30)	60.44 (6.06)	52.79 (4.11)	51.39 (3.08)
	CB10	60.95 (8.06)	62.12 (6.90)	53.08 (4.43)	51.58 (3.18)
	CB11	62.37 (7.83)	62.86 (7.14)	53.89 (5.54)	51.79 (3.55)
w8	CB2	63.45 (7.63)	55.45 (4.84)	61.14 (5.91)	52.20 (3.70)
	CB6	59.70 (8.72)	52.42 (3.99)	58.05 (6.87)	52.31 (3.90)
	CB9	61.38 (8.84)	52.79 (4.11)	59.08 (5.48)	51.52 (2.97)
	CB12	64.15 (6.15)	54.99 (4.92)	62.47 (5.38)	52.21 (3.56)
	CB19	60.82 (8.17)	54.23 (5.20)	60.21 (6.64)	51.82 (3.71)
	CB21	63.22 (7.26)	54.01 (4.39)	61.94 (5.13)	51.85 (3.15)

## Data Availability

The datasets generated and/or analyzed during the current study are available in the Panel Study on Korean Children repository, which is available online at https://panel.kicce.re.kr/pskc/index.do (accessed on 15 April 2021).

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
