# Peer review of "Behavioral Changes in Preschool- and School-Age Korean Children: A Network Analysis"

_children, 2022, doi:10.3390/children9050677_

Round 1

Reviewer 1 Report

This is an interesting study that I enjoyed reading. There are a few suggestions and comments that I have to help clarify things in this article.

Introduction

  1. I felt it was strange that research questions were made regarding age 5 and 7, but the manuscript includes testing the network stability of the other waves of data starting on line 191. I would suggest that if you want to keep these research questions then the results of these other waves should be presented in a different way, maybe as part of the discussion (starting on line 440) or in the introduction to better provide an argument for why the research questions were specifically only including age 5 and 7. Otherwise, it seems as though the research questions were decided after the analysis.

Methods

  1. How did this sample come to be (n=1323)? Please provide comparisons between the analytic sample (n=1323) and the full sample, or attrition analysis.
  2. The cut-offs listed on line 97 suggests the clinical cut-off was used, but in the limitations section it is stated that wasn’t used. Please provide the actual cut-off scores that defined the (sub)clinical group from the non-clinical group. Please define what borderline score was actually used.
  3. Along with the above, please provide more description on these (sub) clinical groups. Are these children higher on particular subscales? It would seem to me that these network analyses that are specific to the clinical groups might be dependent on the types of symptomology these two clinical groups are experiencing more of.
  4. Please report reliabilities of these subscales/scale for this sample (n=1323).

Results

  1. Please provide similar reporting of both (sub)clinical and non-clinical groups. For example, starting on Line 251, central nodes for the subclinical group are reported, but the no-clinical group only has the most central node reported. Please provide the central nodes (EI> .05) for the non-clinical group. Also line 290 reports the non-clinical but not the (sub)clinical central nodes.
  2. For clarification, starting on line 285, is this stating that an additional three pairs had high correlations in the (sub)clinical group compared to the non-clinical group? If so please just add the clarification of “was found in an additional three pairs”.

Discussion

  1. I would suggest that the discussion around the posttraumatic stress problems be less diagnostic and more dimensional in interpretation. These are symptoms and are not listed in the tables as PTSD, as such the use of PTSD in text should be replaced by “posttraumatic stress problems”. The argument can still be made as it is on line 482 of the importance of understanding this dimension for PTSD diagnosis, but as it is currently studied it is not PTSD.
  2. In addition, line 474-478 discusses how age 7 results echo similar longitudinal studies. This is a correlational study and not directly testing longitudinal links of ADHD and PTSD. PTSD is not measured at age 5. The longitudinal analysis is more descriptive of the transitions instead of what behaviors at age 5 influence behaviors at age 7. Therefore, this might be better suited for the importance or conclusions of this study, to allow for a more longitudinal study of posttraumatic stress problems.
  3. In addition to the limitation of the cut-off score, it would also be important to note that these analyses are dependent on the specific behaviors exhibited by these (sub)clinical children (i.e., generalizability). For example, if more children in the (sub)clinical group show higher ADHD symptoms compared to a (sub)clinical group that has higher affective problems.

Author Response

We sincerely appreciate the time spent in reviewing this manuscript and your advice to improve it.

Please, see the attached file, our answers to your queries and comments. We also marked the corrected and revised parts of the text with red. We hope that you find them satisfactory.

Sincerely yours,

Hyo Jeong Jeon

Reviewer 2 Report

This study is important but the clinical significance is not clear. Due to that it has limited interest to a larger audience. I would commend the authors in using public data but would ask them to consider acknowledging the participation of the families that reported the data. 

I would encourage to explain limitations of the CBCL to help with clinical significance especially when you think of network analysis. Kindly consider the wider audience.

Author Response

(The authors gave the same response as above.)
